# Airway and Oral microbiome profiling of SARS-CoV-2 infected asthma and non-asthma cases revealing alterations–A pulmonary microbial investigation

Karthik Sekaran[1], Rinku Polachirakkal Varghese[1], George Priya Doss C.[1]*, Alsamman M. Alsamman[2], Hatem Zayed[3], Achraf El Allali[4]*

1 Vellore Institute of Technology, School of Biosciences and Technology, Vellore, India, 2 Department of Genome Mapping, Molecular Genetics and Genome Mapping Laboratory, Agricultural Genetic Engineering Research Institute, Giza, Egypt, 3 Department of Biomedical Sciences College of Health Sciences, QU Health, Qatar University, Doha, Qatar, 4 African Genome Center, Mohammed VI Polytechnic University, Ben Guerir, Morocco

* Achraf.ELALLALI@um6p.ma (AEA); georgepriyadoss@vit.ac.in (GPDC)

**Data Availability Statement:** All relevant data are within the paper and its Supporting Information files.

## Abstract

New evidence strongly discloses the pathogenesis of host-associated microbiomes in respiratory diseases. The microbiome dysbiosis modulates the lung's behavior and deteriorates the respiratory system's effective functioning. Several exogenous and environmental factors influence the development of asthma and chronic lung disease. The relationship between asthma and microbes is reasonably understood and yet to be investigated for more substantiation. The comorbidities such as SARS-CoV-2 further exacerbate the health condition of the asthma-affected individuals. This study examines the raw 16S rRNA sequencing data collected from the saliva and nasopharyngeal regions of pre-existing asthma (23) and non-asthma patients (82) infected by SARS-CoV-2 acquired from the public database. The experiment is designed in a two-fold pattern, analyzing the associativity between the samples collected from the saliva and nasopharyngeal regions. Later, investigates the microbial pathogenesis, its role in exacerbations of respiratory disease, and deciphering the diagnostic biomarkers of the target condition. LEfSE analysis identified that *Actinobacteriota* and *Pseudomonadota* are enriched in the SARS-CoV-2-non-asthma group and SARS-CoV-2 asthma group of the salivary microbiome, respectively. Random forest algorithm is trained with amplicon sequence variants (ASVs) attained better classification accuracy, ROC scores on nasal (84% and 87%) and saliva datasets (93% and 97.5%). *Rothia mucilaginosa* is less abundant, and *Corynebacterium tuberculostearicum* showed higher abundance in the SARS-CoV-2 asthma group. The increase in *Streptococcus* at the genus level in the SARS-CoV-2-asthma group is evidence of discriminating the subgroups.

**Funding:** The authors received no specific funding for this work.

**Competing interests:** The authors have declared that no competing interests exist.

## Introduction

The SARS-CoV-2 virus has been identified as a causative agent of Coronavirus (COVID-19), leading to over 620 million cases and 6.5 million fatalities globally [1]. COVID-19 can appear as an asymptomatic condition or with symptoms differing from moderate respiratory distress to severe lung damage and pervasive inflammation, eventually causing death. Since the pandemic's beginning, researchers have been studying the relevance of comorbidities in susceptibility to SARS-CoV-2 infection. As viral infections augment the likelihood of asthma exacerbations, patients with chronic lung conditions, such as asthma, were initially designated vulnerable to severe COVID-19 illness [2]. However, limited research has been conducted on understanding the links between SARS-CoV-2 infections and respiratory illnesses such as asthma. The Global Initiative for Asthma (GINA) defines asthma as a diverse chronic condition that significantly impacts the quality of life of individuals of all ages. It is characterized by airflow impediment and persistent airway inflammation affecting more than 300 million individuals globally. Asthma patients have altered airway microbiota linked to an increased vulnerability to severe respiratory illnesses caused by viral infections [3]. Therefore, there is a critical need to systematically evaluate the microbiome of asthma patients during SARS-CoV-2 infection.

The human microbiome is a collection of microbes such as bacteria, archaea, viruses, and eukaryotic that reside within the human body and have been linked to various illnesses [4]. As the microbiome contributes to the enhancement or impairment of immunological and metabolic systems, there is clear evidence that the interaction between microorganisms and humans has a pivotal role in defining the condition of illnesses in the human body [5]. Exploring the host-microbe synergisms and microbiome dysbacteriosis in the human body might aid in diagnosing and developing therapeutics against various diseases. Recent clinical investigations have found an elevated expression of angiotensin-converting enzyme 2 (ACE2), a SARS-CoV-2 receptor, in the respiratory and alimentary tract during COVID-19 infection, indicating dysbacteriosis [6–8].

The advent of lung microbiome research has led to studies comparing the bacterial inhabitants of the lungs in healthy individuals with patients affected with chronic respiratory illnesses such as asthma, chronic obstructive pulmonary disease (COPD), cystic fibrosis, and lung cancer [9–13]. Changes in the respiratory microbiota have been linked to susceptivity in viral infections, particularly in the case of influenza and RSV. Regarding the etiology of COVID-19, changes in the microbiome have been linked to the significant association of respiratory and alimentary tracts [14–16]. Accordingly, a significant decline in the gut bacterial diversity and healthy endosymbionts can be observed in COVID-19 patients with an augmentation in the abundance of pathological microbes such as *Actinomyces*, *Rothia*, *Veillonella*, and *Streptococcus* [17].

Several oropharyngeal microbiome investigations in COVID-19 patients have found predominant bacteria from *Bacteroidota*, *Firmicutes*, and *Proteobacteria* phyla [18,19]. A large cohort study by Gao et al. found that the genera *Halomonas*, *Granulicatella*, *Leptotrichia*, and *Streptococcus* are more predominant in COVID-19-infected patients [20]. The nasopharyngeal microbiota is considered unique from the rest of the respiratory tract. Still, it is particularly significant in COVID-19 because the nasal epithelium may be the initial site of SARS-CoV-2 infection. Studies conducted on the nasopharyngeal microbiome have found *Staphylococcal* and *Corynebacterial* species to be more prominent [21–24]. It is crucial to identify microbiota involved in COVID-19 patients with asthma conditions as asthma patients tend to have disproportionate respiratory microbiota increasing the chances of vulnerability towards viral respiratory tract infections. Respiratory microbiome research of COVID-19 patients with

asthma conditions can help to identify the disease's pathogenesis and severity. Limited investigations have been done on SARS-CoV-2 infection in patients with asthma. The proposed study investigates the microbial dynamics of the SARS-CoV-2 infected pre-existing asthma and non-asthma groups. The statistical evaluation of the samples collected from nasopharyngeal and salivary regions unveiled novel insights into the disease condition.

## Materials and methods

### Data acquisition and study design

"COVID-19 & Asthma" keywords were used to search the NCBI BioProject by applying the filters "human" as the organism and "metagenome" as the study type. The collected samples from the nasopharyngeal and salivary microbiomes of patients with and without pre-existing asthma in response to SARS-CoV-2 infection were sequenced using 16S ribosomal RNA sequencing to create the final dataset (BioProject ID: PRJEB51261). The data were processed on Illumina MiSeq Instrument at Washington University in St. Louis School of Medicine and sequenced using the Amplicon technique. This study intends to evaluate the differences in the nasopharyngeal and salivary microbiomes between COVID-19 patients with and without pre-existing SARS-CoV-2 infection. The dataset files were obtained from the Sequence Read Archive (SRA) database using the SRA toolbox. The samples have been labeled according to the patients and the controls. The initial dataset consists of 111 samples, of which 23 belong to the SARS-CoV-2-asthma category, and 82 are to the SARS-CoV-2-non-asthma groups. Six samples were removed due to inaccurate meta-information provided in the database, reducing the count to 105. The dataset is split into two based on the sample collection region, oral and nasal.

### Bioinformatics processing

The 16S rRNA sequencing data files of nasopharyngeal and saliva samples were examined using Quantitative Insights into Microbial Ecology version 2 (QIIME 2 v. 2022.8) (https://qiime2.org/). The raw FASTQ input data were imported into QIIME 2 artifact using Casava 1.8 paired-end demultiplexed format containing paired-end reads. The summary of the demultiplexed artifact was visualized to check the quality of the reads. The trimming and truncation were performed based on the minimum quality score of 30. After interpretation, they were denoised using the DADA2 process [25]. All the reads attained a quality score of more than 30. The sequences were truncated at 240th base length for both forward and reverse reads, and the trimming started at position 0 for both reads. The forward and reverse reads were merged, and DADA2 filters out chimeras. The lower limit of the sampling depth was identified to retain all the samples. The saliva data had a lower sampling depth of 1112/734, whereas the nasal data showed a sampling depth of 1003/811. The features with very low abundance (10) were removed from the feature table. SILVA database (v. 13_8) pre-clustered the ASVs at a 99% sequence similarity threshold, and the pre-trained Naïve Bayes algorithm was used for taxonomic classification [26]. The DNA sequence was matched to a microbial monotype at the phylum or species level using taxonomy analysis. The sequences belonging to Archaea and Eukaryota at the kingdom level and mitochondria and chloroplast at the phylum level were excluded during the preprocessing. The resultant QIIME artifacts, such as the feature frequency table, taxonomy table, and phylogenetic tree, were further analyzed statistically.

The alpha and beta diversity analyses were performed to evaluate the subgroups' diversity and richness of microbes. Alpha diversity of samples between two groups was calculated with Chao1, Shannon, and Simpson index measures [27]. The beta diversity was conducted using permutational multivariate analysis of variance (PERMANOVA) with Bray-Curtis distance

matrices [28]. The results were plotted using the microeco R package [29]. The overlapping species between the salivary and nasal microbiomes were fetched using the Molbio Tools (https://molbiotools.com/listcompare.php). To find the disparate biomarkers among the classes and subclasses, Kruskal-Wallis (KW) sum-rank test and pairwise Wilcoxon test were performed with an alpha value of 0.05. The logarithmic LDA score of 2.0 identified the discriminative features. The linear discriminant analysis size effect (LEfSe) model outputs a cladogram and bar graph, plotted to visualize the taxonomic features (https://huttenhower.sph.harvard.edu/galaxy/) [30]. A random forest algorithm was employed to identify ASVs with higher importance, trained with 75% of samples, and tested with the remaining 25%.

### Machine learning

The 'Boruta' R package calculates the significant predictors from the frequency table based on the RandomForest model [31]. The parameter maxRuns was set to 300 with a p-value of 0.01 to generate the result.

## Results

To examine the relationship between the SARS-CoV-2-Asthma and SARS-CoV-2-non-Asthma groups using ASVs, 111 16S metagenomes sequenced samples were retrieved from the SARS-CoV-2 asthma study, including 23 SARS-CoV-2-Asthma (21.9%) and 88 SARS-CoV-2-non-Asthma types (78.1%). Six samples were further removed from the study due to insufficient meta-information. The remaining 105 samples were further grouped into nasopharyngeal and saliva samples. A total of 57 saliva samples were studied, of which 16 belong to SARS-CoV-2-Asthma, whereas the remaining 41 belong to the SARS-CoV-2-non-Asthma. Seven nasopharyngeal samples belong to SARS-CoV-2-Asthma and 41 to the SARS-CoV-2-non-Asthma. The baseline characteristics of the samples are described in Table 1.

The ASVs and sequence similarity between different racial groups were plotted as a Venn diagram in Fig 1.

### Microbiome composition in the disease groups

After the quality filtering process, there were 120,773 reads for saliva samples, with a mean of 2,118 reads per sample. The total number of reads for nasopharyngeal samples was 63,515, with a mean of 1,321 reads per sample. A total of 2,840 and 1,366 ASVs were found for salivary and nasopharyngeal samples after a 99% similarity level was identified on sequences with the SILVA database. The mean relative taxon abundancies were computed for the nasopharyngeal and salivary microbiome data from SARS-CoV-2-Asthma and SARS-CoV-2-non-Asthma groups. The top 5 bacteria at the phylum level in the salivary and nasopharyngeal microbiome

**Table 1. Baseline characteristics of sample groups.**

| Sample Characteristics | Oropharyngeal Samples | Nasopharyngeal Samples |
|---|---|---|
| No. of individuals | 57 | 48 |
| SARS-CoV-2-Asthma group | 16 | 7 |
| SARS-CoV-2-non-Asthma group | 41 | 41 |
| Mean age | 54.01228 | 64.28125 |
| Gender(M/F) | 20/37 | 20/28 |
| Ethnicity | African-American: 38 | African-American: 37 |
| | Asian: 1 | Asian: 1 |
| | White: 18 | White: 10 |

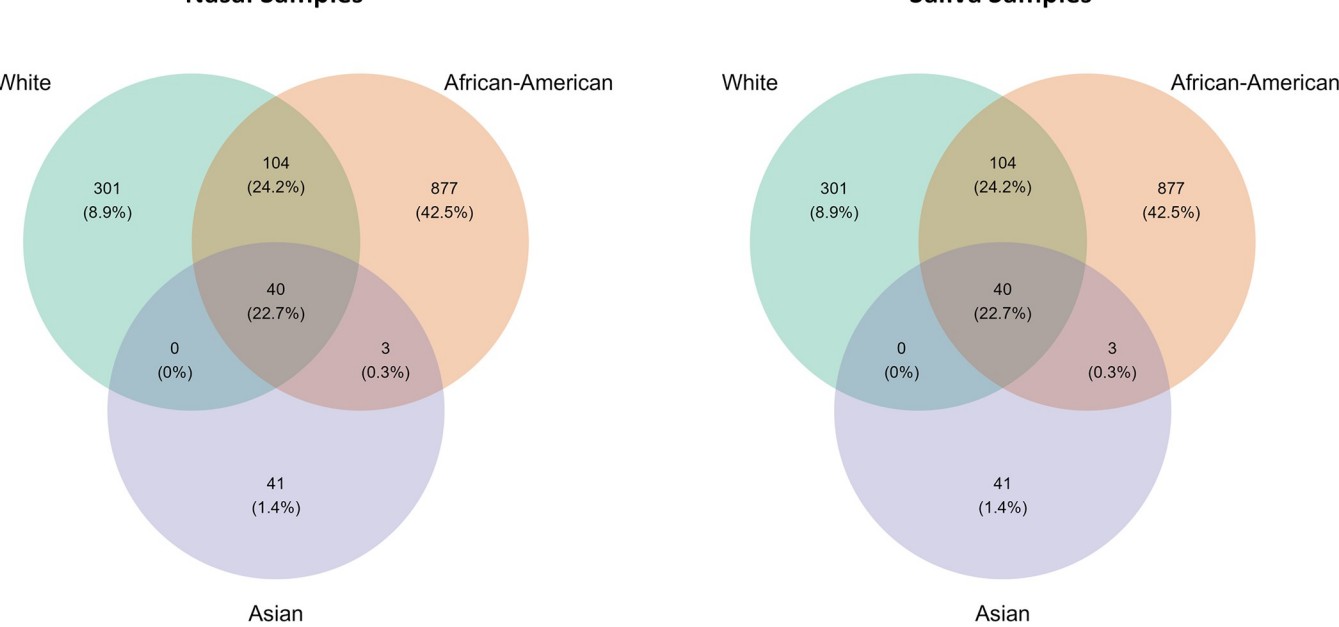

**Fig 1. The Venn diagram represents the unique and shared ASVs between different race groups.** The numerical value indicates the number of ASVs of the corresponding race, and the percentage is the sequence similarity (Nasal Dataset–Left; Saliva Dataset—Right).

were *Firmicutes*, *Actinobacteria*, *Proteobacteria*, *Bacteroidota*, and *Fusobacteria*, with *Firmicutes* being the most abundant phyla both sample-wise (Fig 2) and gender-race wise (Fig 3) representing over 80% of the total phyla within the groups. In salivary and nasopharyngeal microbiomes, there were no significant differences in abundancies between SARS-CoV-2-Asthma and SARS-CoV-2-non-Asthma groups.

*Bacilli* and *Actinobacteria* classes showed a higher predominance in nasal and salivary microbiomes in SARS-CoV-2-Asthma and SARS-CoV-2-non-Asthma groups (Fig 4). *Burkholderia-Caballeronia-Paraburkholderia*, *Pseudomonas*, *Cutibacterium*, *Staphylococcus*, and *Corynebacterium* were found at the genus level to be predominant in the nasopharyngeal microbiome. *Corynebacterium*, *Pseudomonas*, and *Burkholderia-Caballeronia-Paraburkholderia* showed higher abundance in SARS-CoV-2-Asthma group. Conversely, it showed lesser abundance in the SARS-CoV-2-non-Asthma group (Fig 5). *Actinomyces*, *Veillonella*, *Prevotella*, *Rothia*, and *Streptococcus*, showed more prevalence in the salivary microbiome. *Actinomyces* and *Veillonella* showed higher abundance in SARS-CoV-2-Asthma and lesser abundance in the SARS-CoV-2-non-Asthma group, whereas *Rothia* was found to be more prevalent in SARS-CoV-2-non-Asthma and vice versa in SARS-CoV-2-Asthma.

At the species level, *Corynebacterium psuedodiptheriticum* and *Corynebacterium tuberculostearicum* showed higher abundance in the SARS-CoV-2-Asthma group in the nasal microbiome. In contrast, in the salivary microbiome, *Rothia mucilaginosa* showed a downtrend in the SARS-CoV-2-Asthma group and a higher abundance in the SARS-CoV-2-non-Asthma group (Fig 6). The heatmap depicts the difference among the racial groups such as African-American, Asian, and White at the class level (Fig 7). *Streptococcus* is found predominantly in both microbiomes, followed by *Rothia*, *Prevotella*, *Veilonella*, and *Actinomyces* in the salivary microbiome, whereas *Staphylococcus*, *Corynebacterium*, *Cutibacterium*, and *Dolosigranulum were found* in the nasopharyngeal microbiome.

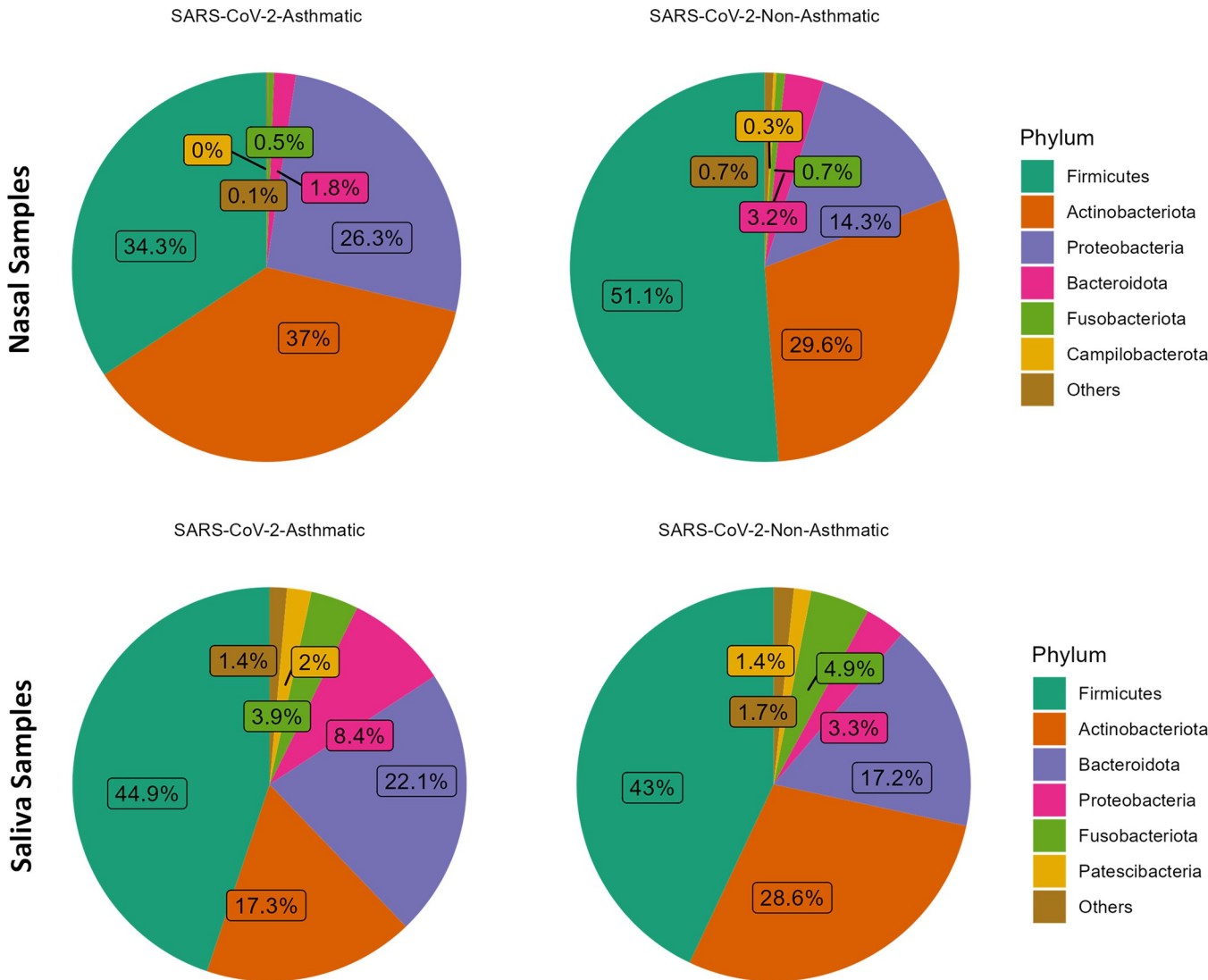

**Fig 2. Pie chart with the distribution of microbiome at phylum level displaying top abundance (Nasal Dataset–Top; Saliva Dataset—Bottom).**

## Alpha, beta diversity estimation

The various features of the groups (SARS-CoV-2-Asthma and SARS-CoV-2-non-Asthma) from salivary and nasopharyngeal microbiomes were described using the Chao1, Shannon, and Simpson indexes. The Chao1 index illustrates the species richness within the sample by quantifying the number of ASVs. In contrast, the Shannon and Simpson index can explain community diversity (species richness and diversity). Alpha diversity measures the species heterogeneity within the samples; the Shannon and Simpson indices can be computed using the species from the ASVs and their abundance.

In the nasopharyngeal microbiome, the Chao1 measure showed a considerably higher index for female patients in SARS-CoV-2-Asthma compared to the SARS-CoV-2-non-Asthma group. In contrast, in male patients, a higher index was seen in SARS-CoV-2-non-Asthma than in the SARS-CoV-2-Asthma group. As per the findings, the species richness was substantially higher in SARS-CoV-2-Asthma female patients and SARS-CoV-2-non-Asthma male

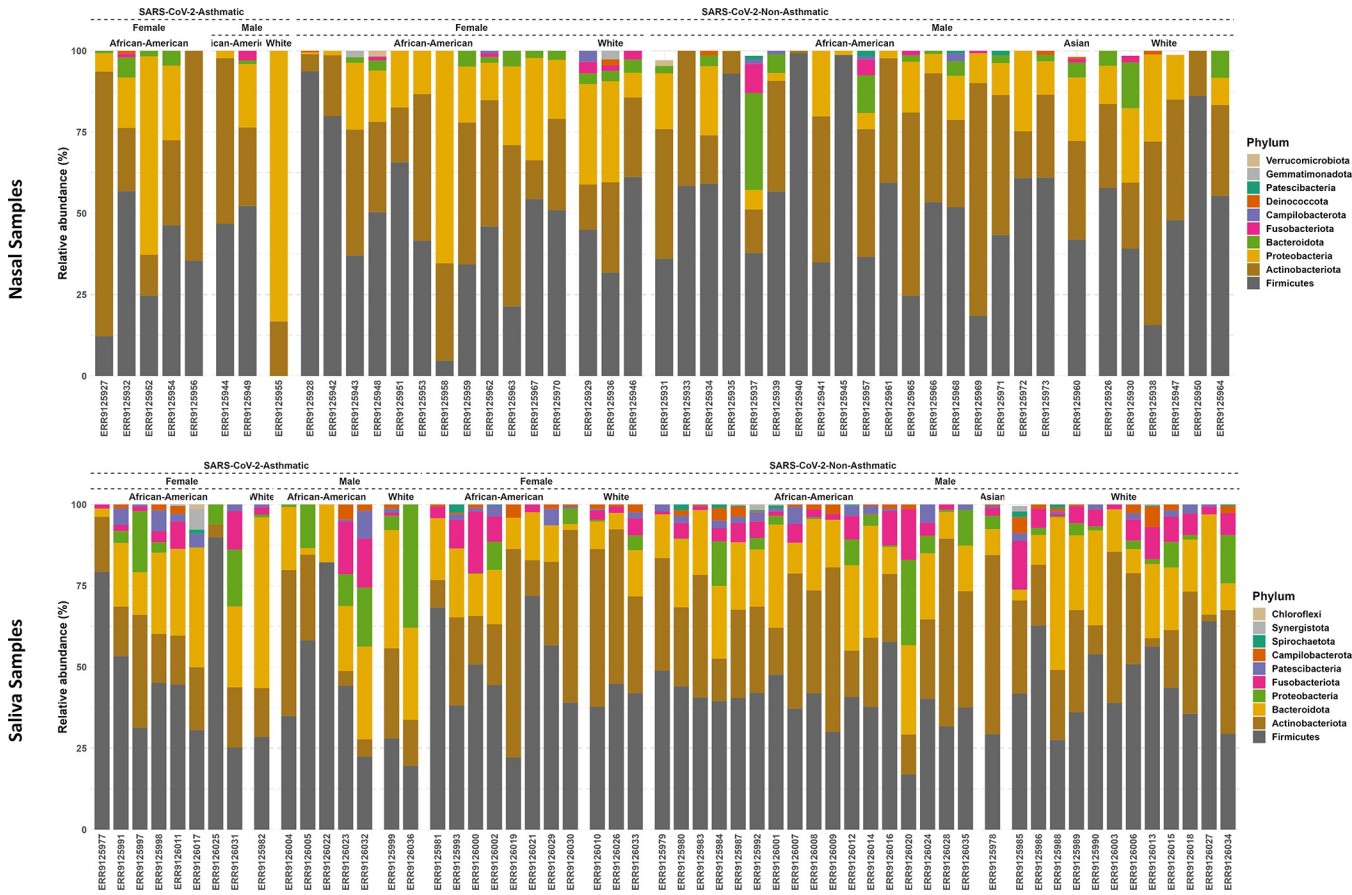

**Fig 3. Bar plots of relative frequency grouped by gender and race at phylum level representing the most abundant taxa.** The distinctive bars represent each ASVs on the sequence at a relative abundance for SARS-CoV-2-asthma and SAR-CoV-2-non-asthma groups (Nasal Dataset–Top; Saliva Dataset—Bottom).

patients. In the Shannon index, higher indexes were shown by male and female patients belonging to the SARS-CoV-2-non-Asthma group. In the Simpson index, a higher index was observed in female patients of the SARS-CoV-2-Asthma group and male patients of the SARS-CoV-2-non-asthma group.

The Chao1 measure in the salivary microbiome also showed a higher index for female patients in SARS-CoV-2-Asthma, whereas the higher index in the male patient was seen in the SARS-CoV-2-non-Asthma group. In Shannon and Simpson measure, higher indexes were shown in female patients of the SARS-CoV-2 Asthma group, and male patients in the SARS-CoV-2-non-Asthma group showed higher indexes (Fig 8). Principal coordinates analysis (PCOA) demonstrates variations among samples to better highlight variances in species richness. The species makeup of the two samples is comparable if the two samples are near to one another. PCOA analyses based on the Bray-Curtis index were computed to determine the variance between the nasopharyngeal and salivary microbiome. The beta diversity measured by Bray Curtis distance method showed a minimal difference of microbial communities in nasopharyngeal (p-value: 0.85, $R^2$: 0.016, F-value: 0.75) and salivary microbiome (p-value: 0.4, $R^2$: 0.017, F-value: 1.008) (Fig 9). The Molbio tool identified 31 shared species between the nasopharyngeal and salivary microbiome, of which bacteria belonging to *Streptococcus*, *Prevotella*, and *Lactobacillus* genera were found to be more predominant (S1 Table).

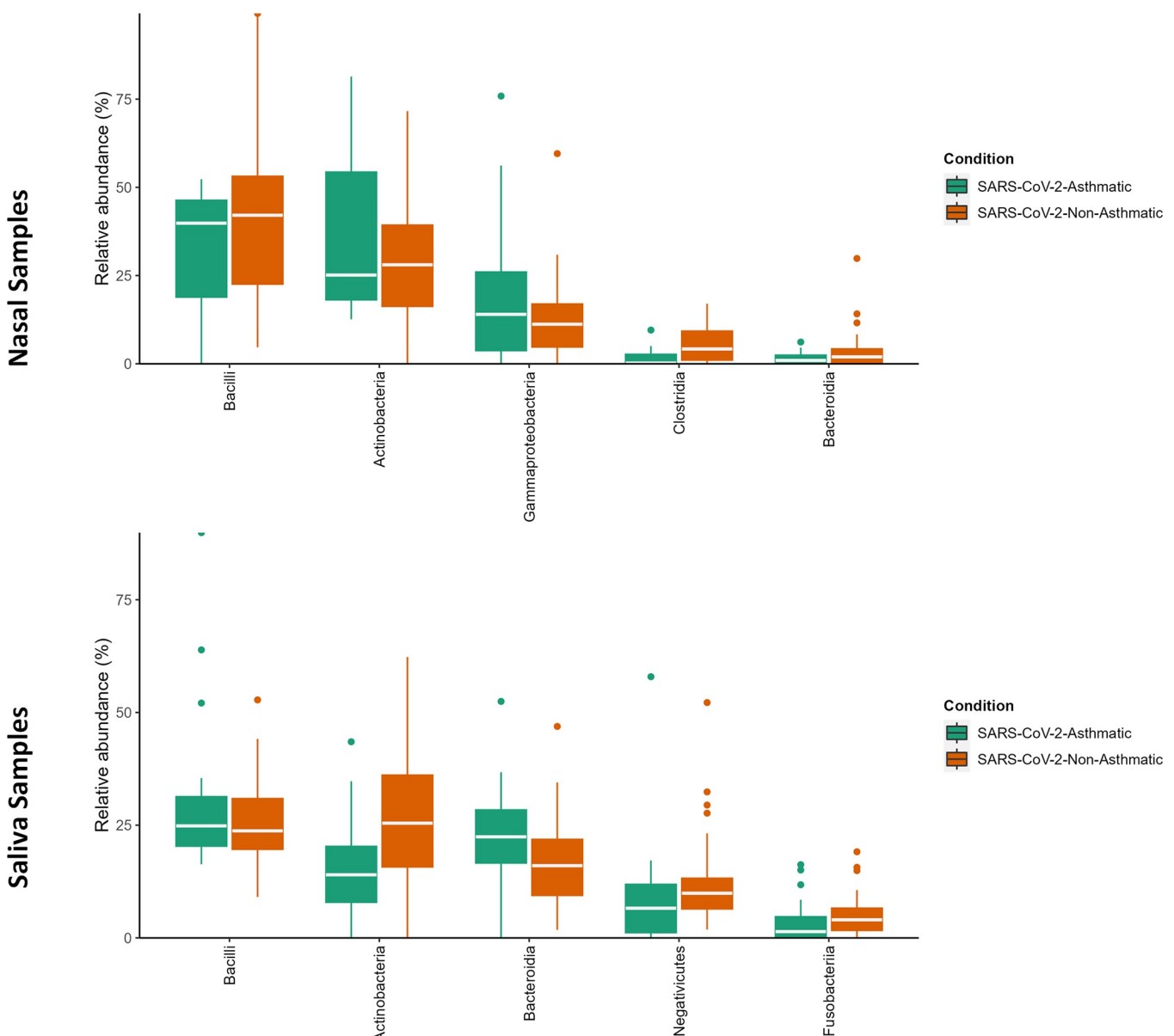

**Fig 4. Boxplot of abundance at the genus level.** The upper and lower bound quartiles represent the 25[th] and 75[th] percentiles; the lines denote the maximum and minimum values, whereas the dots depict the outliers. The SARS-CoV-2-asthma and SAR-CoV-2-Non-asthma conditions have been shown in green and orange colors. (Nasal Dataset–Top; Saliva Dataset—Bottom).

### Linear discriminant effect size analysis

LEfSe assessment identifies the microbiome abundance in the SARS-CoV-2 Asthma and SARS-CoV-2-non-Asthma groups from the nasopharyngeal and salivary microbiome. The LEfSe profiling shows variations within the nasopharyngeal and salivary microbiome at several taxonomic levels with a threshold of LDA score 2.0 (Fig 10). The cladogram revealed variations between SARS-CoV-2-Asthma and SARS-CoV-2-non-Asthma groups in the nasal microbiome *Peptostreptococcales-Tissierellales* showed a substantial increase in SARS-CoV-2-non-Asthma whereas *Rhizobales* showed a significant abundance in SARS-CoV-2-Asthma group. While *Actinobacteriota* in the salivary microbiome of the SARS-CoV-2-Asthma group had

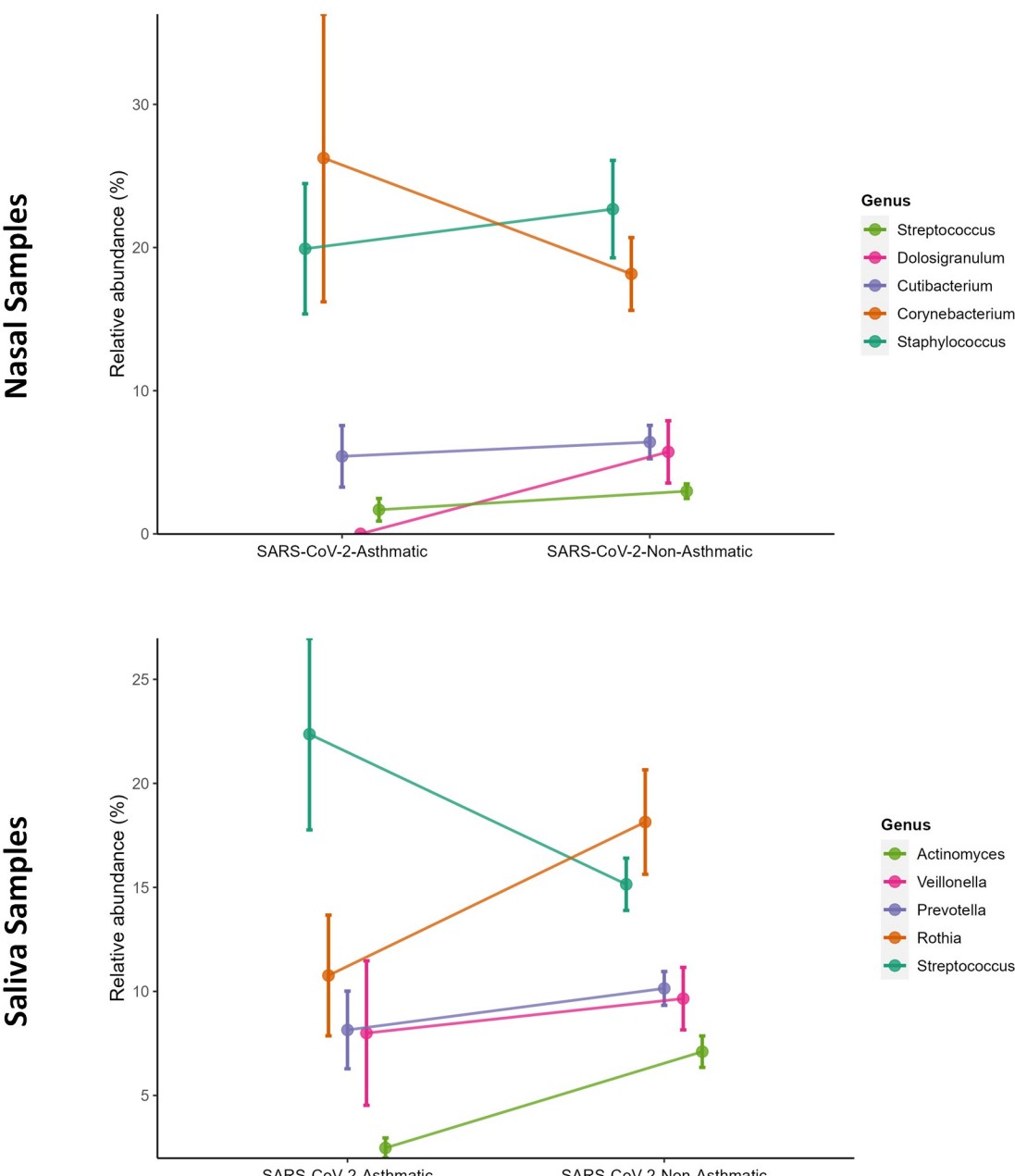

**Fig 5.** The line plot representing the taxonomic abundances at genus level between SARS-CoV-2-asthma and SARS-CoV-2-non-asthma groups of nasal (top) and saliva datasets (bottom). The taxa are color-coded as per the genus-level assignments.

increased significantly, *Pseudomonadota has* shown significant abundance in the SARS-CoV-2-non-Asthma group.

## Machine learning model performances

In total, eight features were selected by the model for saliva and two for nasal. Further, the random forest classifier is trained on the reduced feature set with a maximum tree size of 500. The data was split into 75%-25% for the training and testing process. The classification accuracy attained by the model on the salivary microbiome is 93% with an AUC score of 97.5%, and on

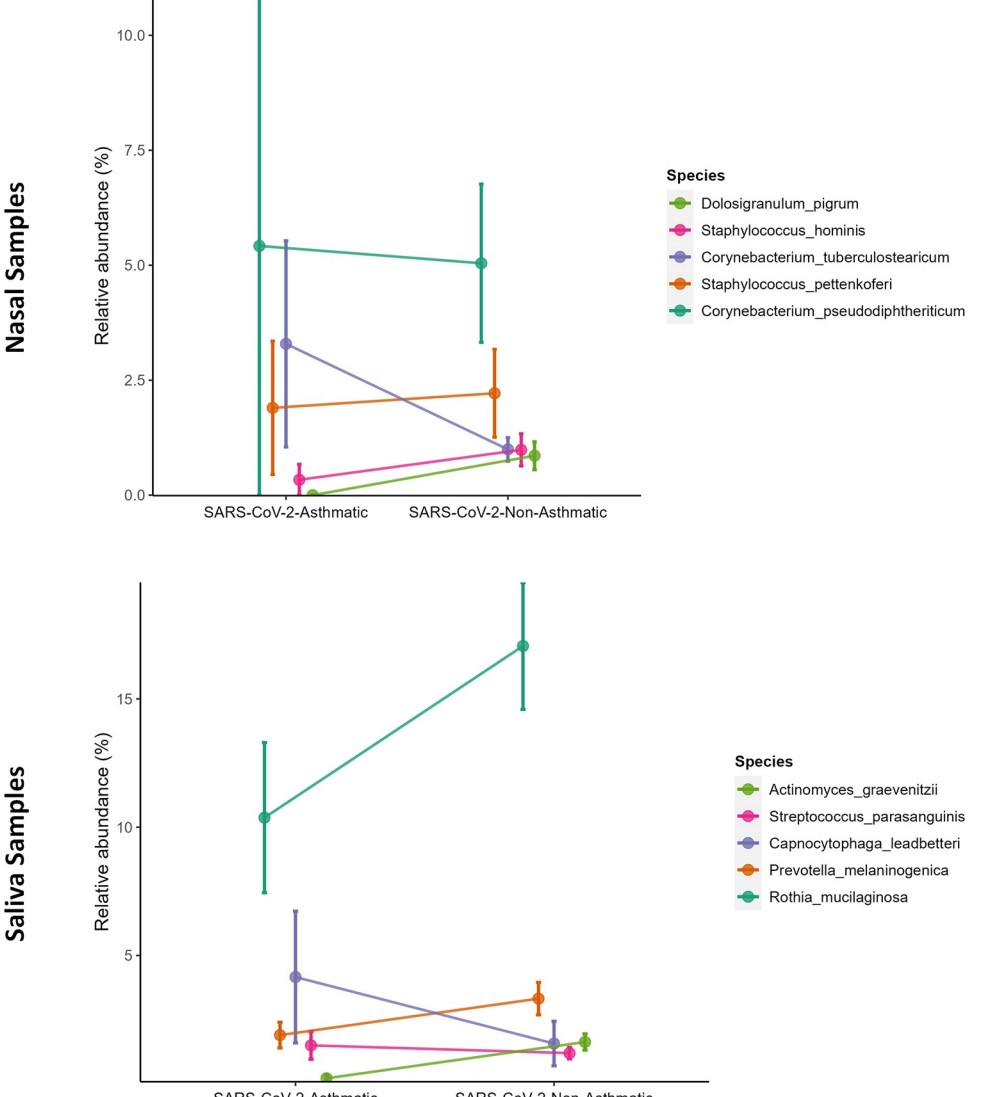

**Fig 6.** The line plot depicts the significant difference between SARS-CoV-2-asthma and SARS-CoV-2-non-asthma groups of nasal (top) and saliva datasets (bottom). The taxa are color-coded as per the species-level assignments.

the nasal microbiome is 84% with an AUC score of 87%, respectively. The AUC plots for the nasal and saliva microbiomes are represented in Fig 11.

## Discussion

The oropharyngeal and nasopharyngeal regions are significant entry points and SARS-CoV-2 reservoirs [32]. Interactions between the SARS-CoV-2 virus and commensal microorganisms within the pharynx region are important in mediating the viral loads and host immunological responses [19,32]. It is pivotal to identify microbial factors in COVID-19 patients with and without pre-existing asthma as an imbalance of microbial community in patients with asthma has been characterized as a factor for asthma-associated susceptibility to viral infections [33–35]. As limited investigations have been done on SAR-CoV-2 infection in patients with prior asthma conditions, our study aims to offer a novel by investigating the microbial pattern of SARS-CoV-2 infected individuals with pre-existing asthma and non-asthma groups.

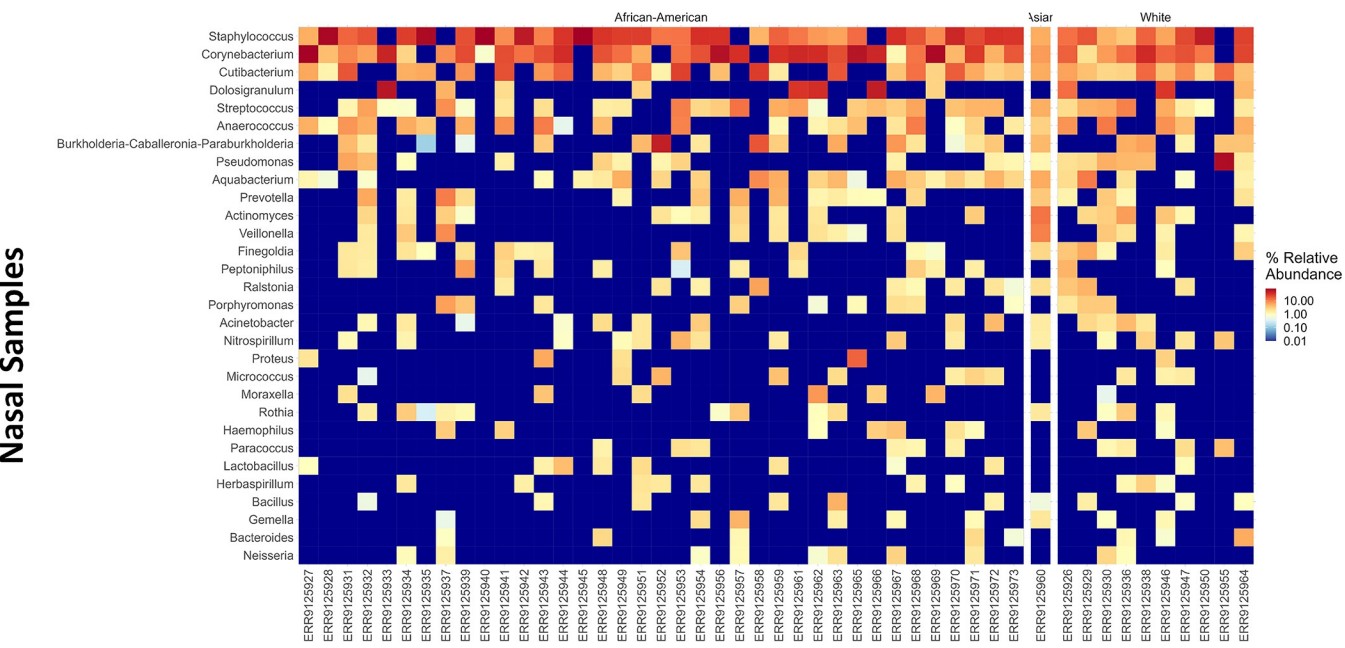

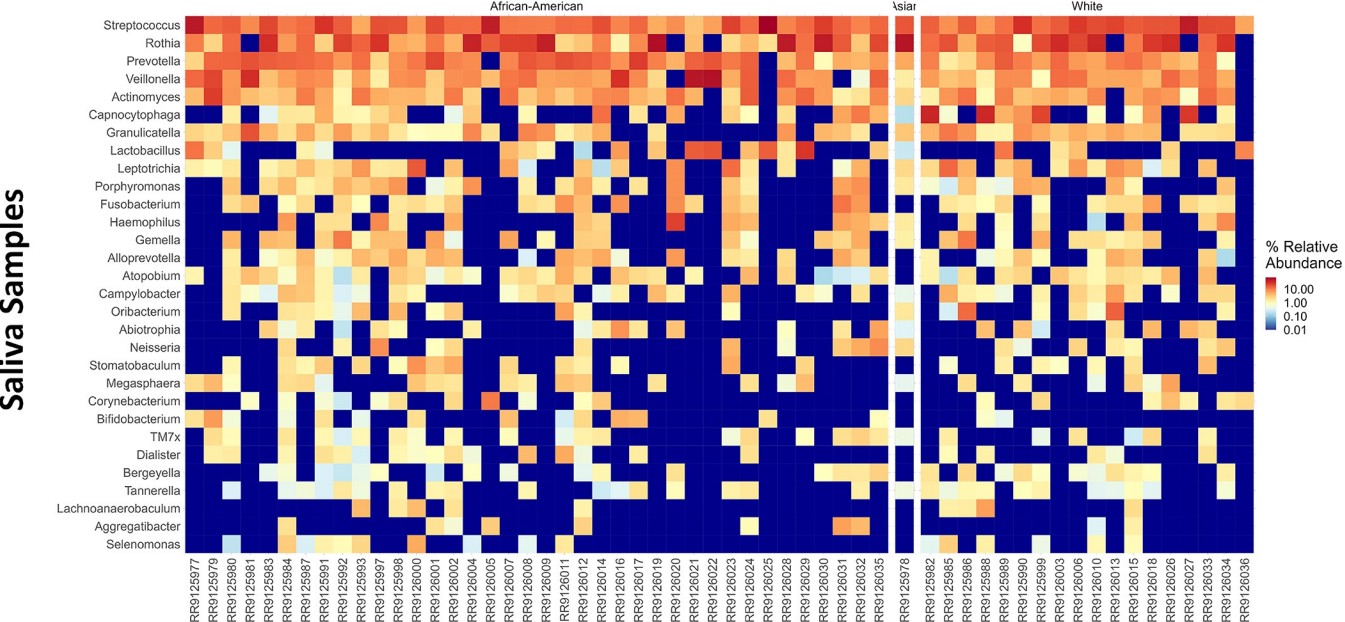

**Fig 7.** Heatmap visualizes the difference between African-American, Asian, and White races at the class level (Nasal Dataset–Left; Saliva Dataset—Right). Each square in the heatmap represents the given ASVs relative abundance; the orange color's intensity correlates with the class-level abundance.

Firmicutes, *Actinobacteria*, *Bacteroidota*, and *Proteobacteria*, were found to be the most abundant at the phylum level in both nasal and salivary microbiomes. There was a slight variation in the gender and race-wise abundance at the phylum level. Conversely, the pattern variation is observed at the genus level compared to the asthma condition. Corynebacterium is higher in the SARS-CoV-2-Asthma group than non-asthma, followed by *Staphylococcus*, *Cutibacterium*, *Pseudomonas*, and *Burkholderia-Caballeronia-Paraburkholderia* on the nasopharyngeal microbiome. *Streptococcus* shows a more significant difference between the

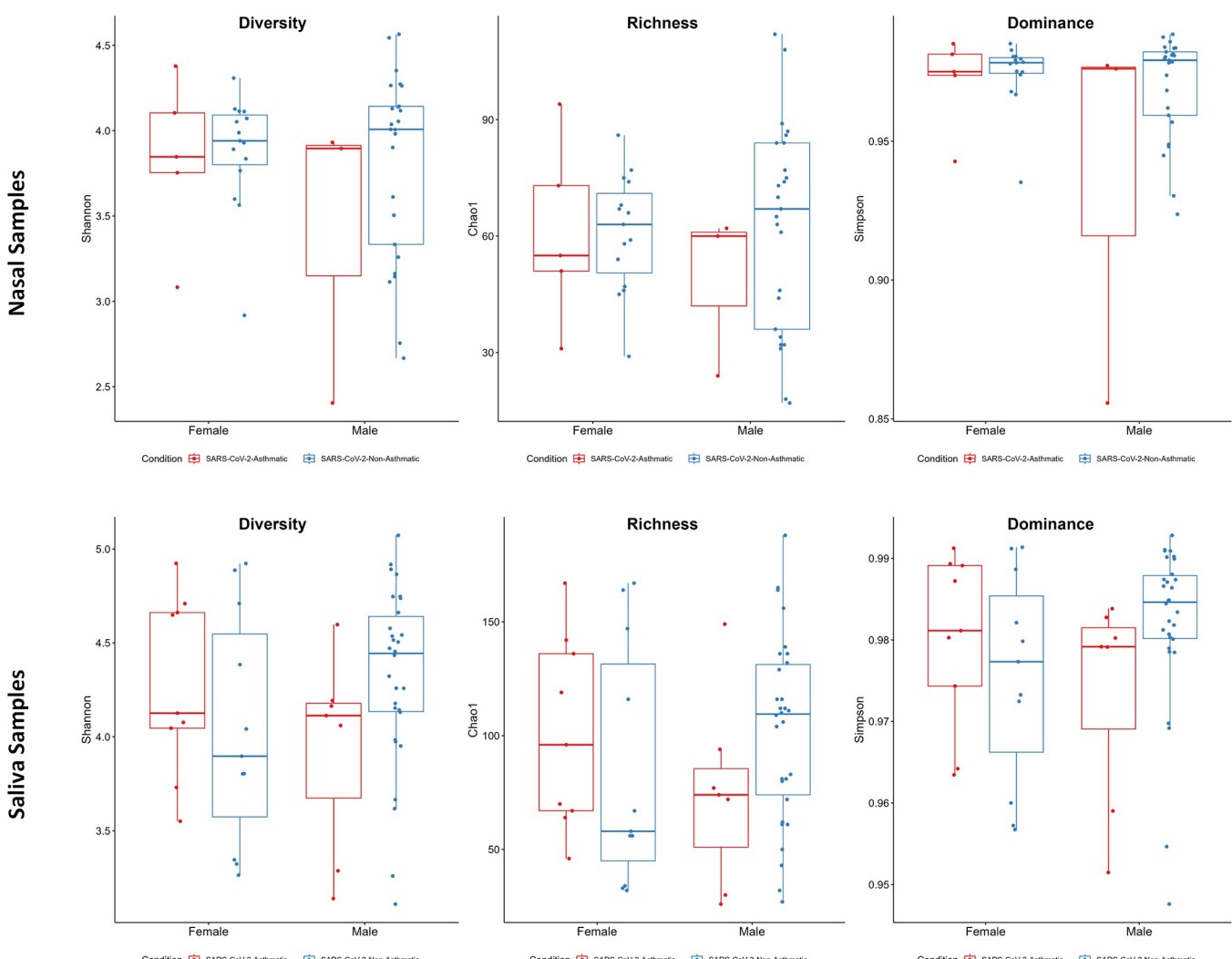

**Fig 8. Box plots of Alpha diversity indices.** The Shannon and Simpson indices depict the ASV diversity, whereas the Choa 1 represents the ASV abundance in the SARS-CoV-2-asthma and SARS-CoV-2-non-asthma conditions at gender level groupings. (Nasal Dataset–Top; Saliva Dataset—Bottom). The boxes show the interquartile range between the 25th and 75th percentile, whereas horizontal lines and colored dots represent the median and outliers. The SARS-CoV-2-asthma and SARS-CoV-2-Non-asthma groups were shown in green and orange colors. (Nasal Dataset–Top; Saliva Dataset—Bottom).

SARS-CoV-2-asthma and SARS-CoV-2-non-Asthma groups in the salivary microbiome, followed by *Rothia*, *Prevotella*, *Veillonella*, and *Actinomyces*. Multiple studies corroborate the increase in severity with the presence of *Streptococcus* [36–39]. To further evaluate the pathogenic microbes, we compared the prevalence of bacterial genera between the SARS-CoV-2-Asthma and SARS-CoV-2-non-Asthma groups. Surprisingly, none of the top 5 microbes at the genus level is matching between the two microbiomes. The relative abundance at the class level identified *Bacilli* and *Actinobacteria* commonly in nasopharyngeal and salivary microbiomes. However, both classes were found to have a higher abundance in the nasopharyngeal microbiome than the salivary microbiome. Bacteroidia shows lesser prevalence in the nasopharyngeal microbiome but is ranked as the third most abundant bacteria in the salivary microbiome (Fig 4).

At the species level, a significant decline in an anti-inflammatory bacterium, *Rothia Mucilaginosa*, is observed in the SARS-CoV-2-Asthma group (Fig 6) in the salivary microbiome. *Corynebacterium tuberculostearicum*, a pathogenic respiratory species, is abundant in the

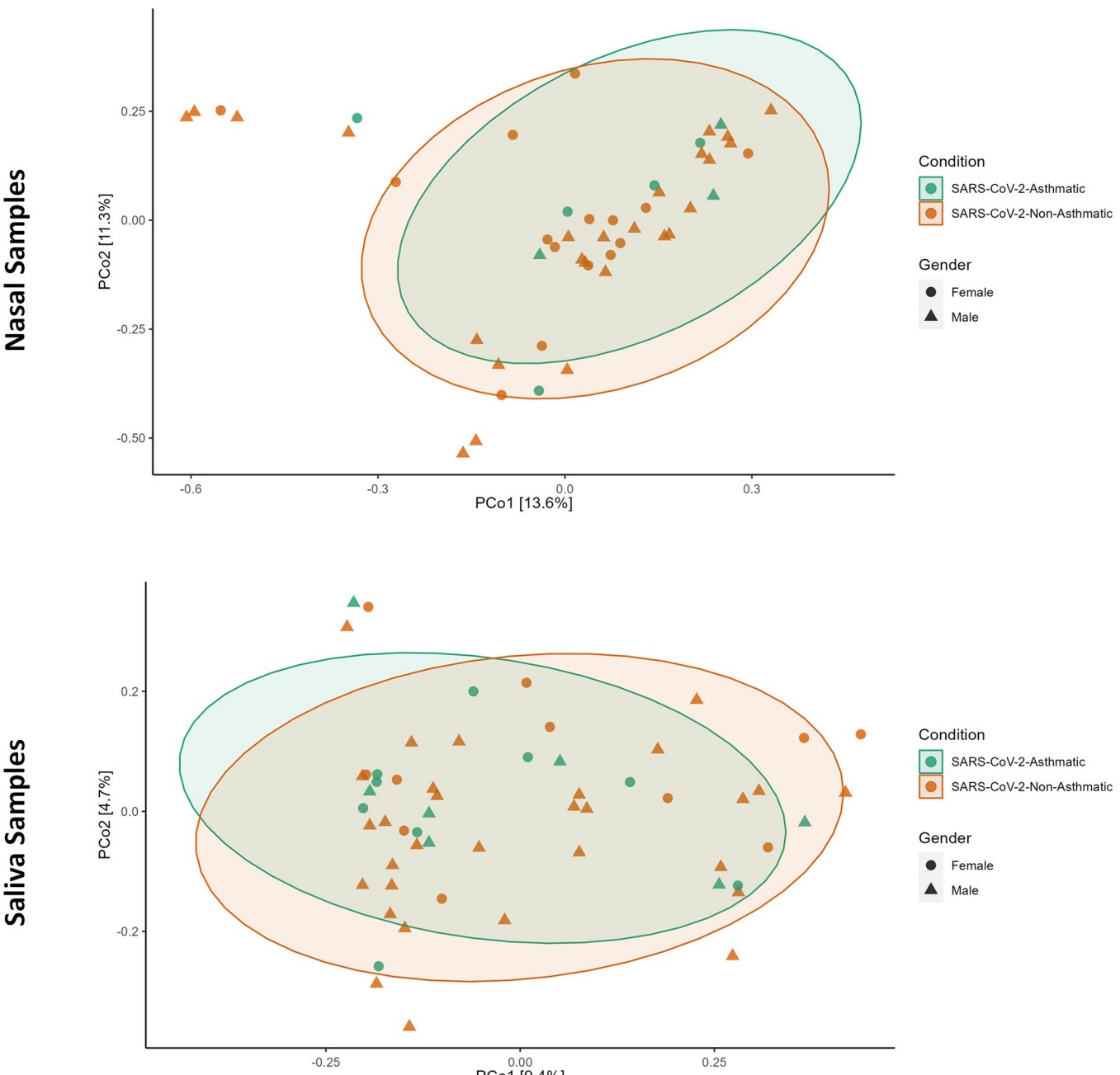

**Fig 9. PCOA plots of Beta diversity based on the Bray-Curtis distance metric.** The ellipses represent SARS-CoV-2-asthma (green) and SARS-CoV-2-Non-asthma (orange) conditions (Nasal Dataset–Top; Saliva Dataset—Bottom).

SARS-CoV-2-Asthma group on the nasopharyngeal microbiome [40,41]. The diversity and richness analysis found significant variation between the nasopharyngeal and salivary microbiomes. The beta diversity evaluation using the Bray-Curtis method identified the p-value scores of 0.85 and 0.4, respectively. Further, the alpha diversity calculated with Shannon, Chao1, and Simpson method represented as a boxplot visualizes substantial variation between the two subgroups, categorized by gender information. The LEfSe plot represents the differential microbial features explaining the variation between groups.

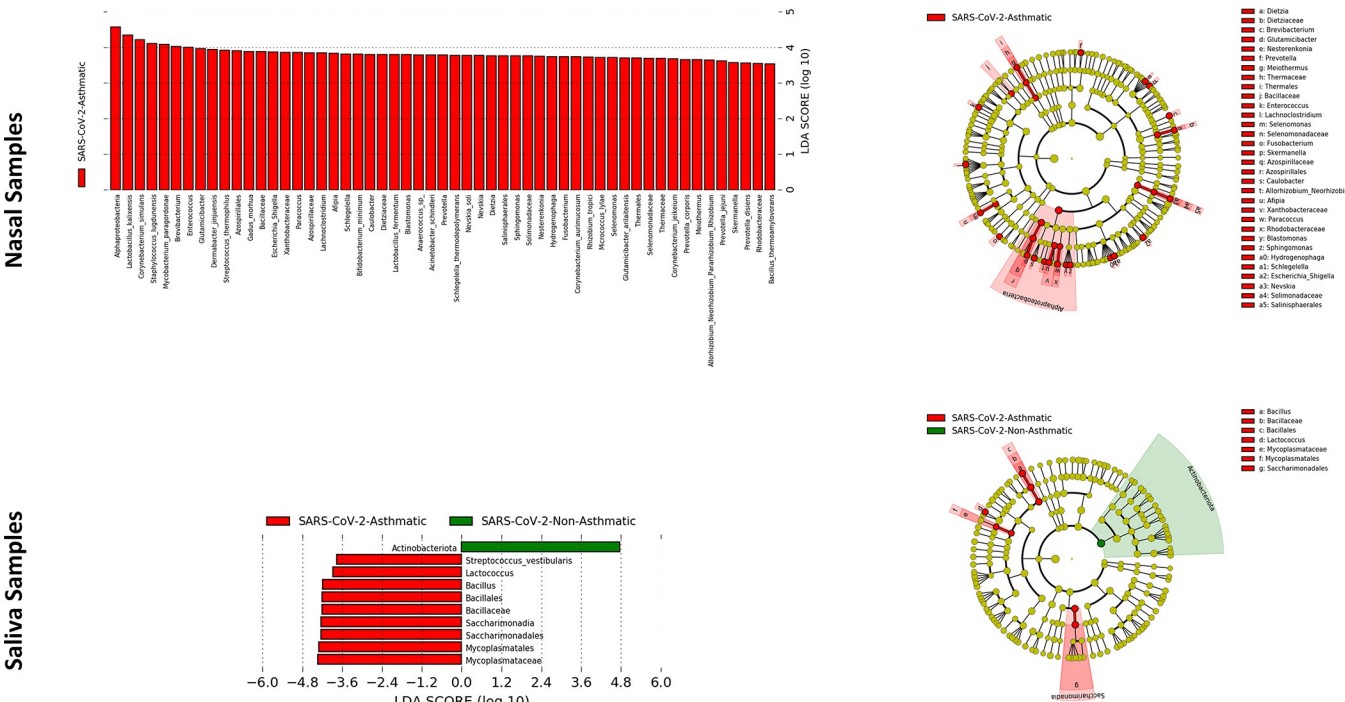

**Fig 10. LEfSe analysis of oropharyngeal and nasopharyngeal microbiome at LDA threshold 2.0 showing differentially abundant taxa between subgroups.**
The barplot shows the enriched bacteria that are associated with SARS-CoV-2-asthma (red) and SARS-CoV-2-Non-asthma (green) groups [Left–(Nasal Dataset–Top; Saliva Dataset—Bottom)] and phylogenetic cladogram plot illustrates the differentially abundant taxa at various taxonomic levels for SARS-CoV-2-asthma (red) and SARS-CoV-2-Non-asthma (green) groups [Right–(Nasal Dataset–Top; Saliva Dataset—Bottom)].

With the threshold LDA scores of 2.0, two crucial markers are *Actinobacteriota* in the SARS-CoV-2-asthma group and the *Pseudomonadota* in the SARS-CoV-2-non-Asthma group at the phylum level is enriched, having differential abundance in the salivary microbiome. The *Peptostreptococcales-Tissierellales* in the SARS-CoV-2-non-Asthma group is enriched, whereas *Rhizobiales* in *the* SARS-CoV-2-Asthma group at order level shows differential abundance in the nasal microbiome. Machine learning algorithms are capable of deciphering profound patterns from complex data structures. In biological studies, identifying informative inferences becomes difficult due to the vague and arbitrary representation of the data. Feature selection techniques eliminate irrelevant attribute information from the data using statistical methods. This study found eight and two most discriminative features (i.e., ASVs) from the frequency table on nasal and saliva microbiomes by calculating the feature importance using a random forest algorithm. The reduced feature set is also trained using the same model with variable parameters to find the performance of the two microbiome sets. The classification accuracy and AUC score on the salivary microbiome are 93% and 97.5%; on the nasopharyngeal microbiome, they are 84% and 87%, respectively. The scores exhibit better discrimination between the classes of subgroups.

## Conclusions

Understanding microbial dynamics in clinical cases is significant in unraveling pathogenic events; thereby, treatment strategies can be improved. Recent studies nominate the microbiome as an authoritative biomarker for studying disease patterns. The abundance of *Streptococcus* at genus level in SARS-CoV-2-asthma samples on salivary microbiome strengthens the proof of an impact on the host respiratory condition with microbial dysbiosis. The decline of

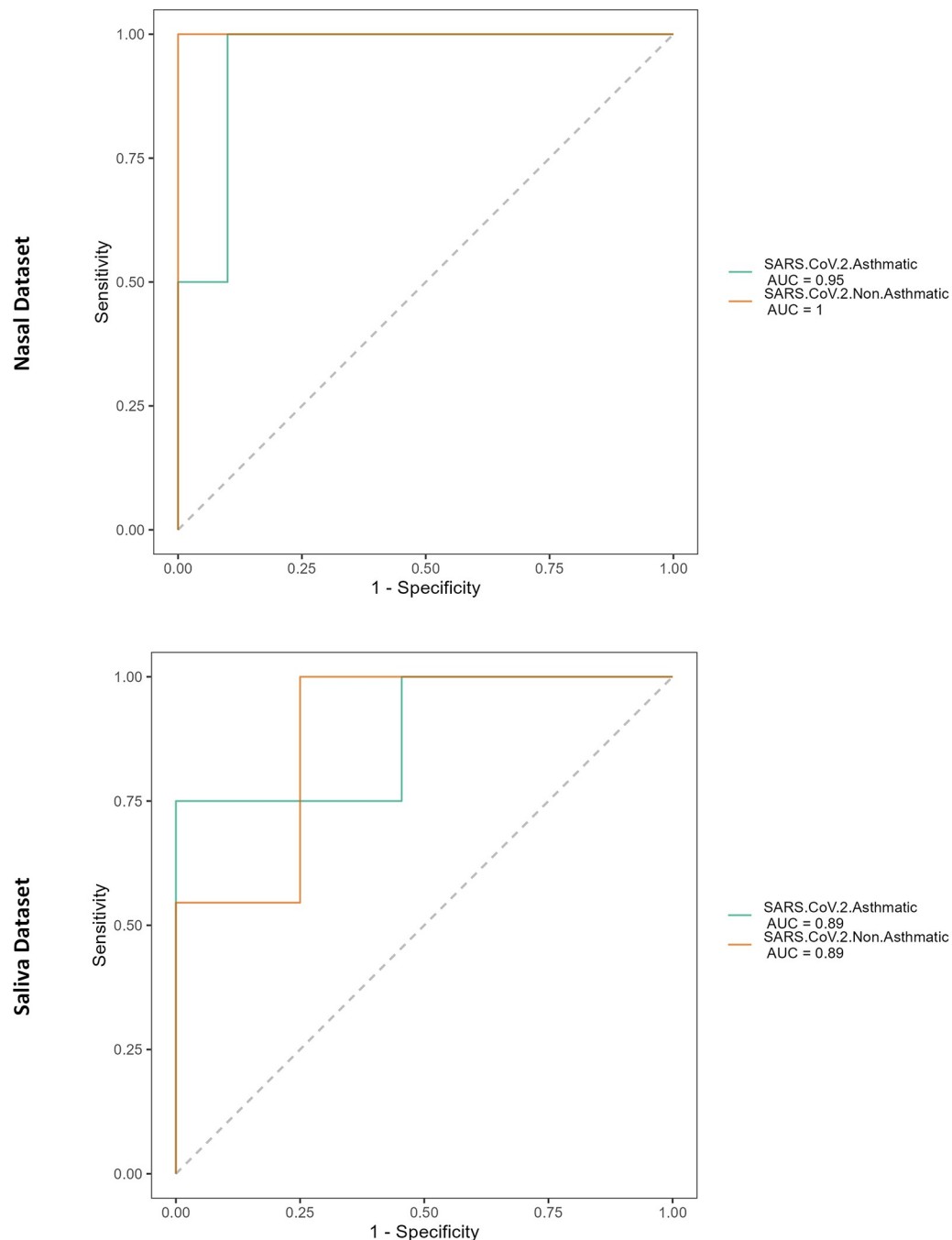

**Fig 11. The Area Under the Receiver Operational Characteristic (AU-ROC) curve for SARS-CoV-2-asthmac (green) and SARS-CoV-2-Non-asthma (orange) groups based on microbial composition.** (Nasal Dataset–Top; Saliva Dataset—Bottom).

*Rothia mucilaginosa* and increase of *Corynebacterium tuberculostearicum* in the SARS-CoV-2-asthma group of the salivary and nasopharyngeal microbiome in both genus and species level is the shred of evidence observed during the experiment. These shreds of evidence advance our understanding of the association between SARS-CoV-2 and asthma and pave the way for future use of the microbiome as diagnostic biomarkers and therapeutic approaches.

## Limitations of the present study

This study has some potential limitations, addressed in brief. The study was not designed to compare disease and control samples due to a lack of data availability on uninfected controls. However, it is intended to better understand the disease etiology by characterizing the microbial dysbiosis between SARS-CoV-2-asthma and SARS-CoV-2-non-asthma groups. The dataset inherently contains a smaller sample size of the SARS-CoV-2-asthma group than the SARS-CoV-2-non-asthma group, which could result in a slight bias. Performing a comparative analysis across different datasets remains challenging without evidence from the matching studies.

## Supporting information

**S1 Table. Overlapping species.**
(XLSX)

## Acknowledgments

The authors would like to thank the authorities of the Vellore Institute of Technology, India, and the University of Mohammed VI Polytechnic, Morocco for providing the necessary support in completing the manuscript. The authors Karthik Sekaran and Rinku Polachirakkal Varghese acknowledge the Indian Council of Medical Research (ICMR), for their fellowships (No. BMI/12(13)/2021, ID No: 2021–6359 & No. VIR/COVID-19/31/2021/ECD-I, ID. NO: 2021–5570).

## Author Contributions

**Conceptualization:** Karthik Sekaran, Rinku Polachirakkal Varghese, George Priya Doss C., Alsamman M. Alsamman, Hatem Zayed, Achraf El Allali.

**Data curation:** Karthik Sekaran, Rinku Polachirakkal Varghese, George Priya Doss C.

**Investigation:** Karthik Sekaran, Rinku Polachirakkal Varghese, George Priya Doss C., Achraf El Allali.

**Supervision:** George Priya Doss C., Alsamman M. Alsamman, Hatem Zayed, Achraf El Allali.

**Writing – original draft:** Karthik Sekaran, Rinku Polachirakkal Varghese.

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
