## [Decision Letter · Decision Letter 0]

14 Jun 2023

PONE-D-23-05664Airway and Oral Microbiome Profiling of SARS-CoV-2 Infected Asthmatic and Non-Asthmatic Cases Revealing Alterations – A Pulmonary Microbial InvestigationPLOS ONE

Dear Dr. El Allali,

Thank you for submitting your manuscript to PLOS ONE. After careful consideration, we feel that it has merit but does not fully meet PLOS ONE’s publication criteria as it currently stands. Therefore, we invite you to submit a revised version of the manuscript that addresses the points raised during the review process.

First, I would like to apologize again for the length of time required to review your manuscript. I have now heard from two reviewers. Reviewer 1 recommended a major revision. The majority of their comments center on the quality of the writing in the manuscript, which I believe can be improved. Reviewer 2 recommended to reject. Their major point is that the study lacks a control group - asthmatics without a SARS-CoV2 infection. I agree that this is an important control group that could help to bolster the findings of the manuscript. If the authors believe that they can address the points raised by both reviewers, I would be happy to consider a revised version.

We look forward to receiving your revised manuscript.

Kind regards,

Robert P Smith

Academic Editor

PLOS ONE

Journal Requirements:

Reviewers' comments:

Reviewer's Responses to Questions

**Comments to the Author**

1. Is the manuscript technically sound, and do the data support the conclusions?

Reviewer #1: Partly

Reviewer #2: No

2. Has the statistical analysis been performed appropriately and rigorously? 

Reviewer #1: Yes

Reviewer #2: Yes

3. Have the authors made all data underlying the findings in their manuscript fully available?

Reviewer #1: Yes

Reviewer #2: Yes

4. Is the manuscript presented in an intelligible fashion and written in standard English?

Reviewer #1: No

Reviewer #2: No

5. Review Comments to the Author

Reviewer #1: The authors present a study analyzing the compositiond and potential role of the oro-pharyngeal microbiome in the increased susceptibility of asthmatic subjects to COVID19.

The topic is interesting and sound, however I have several major criticisms.

First, the paper is quite difficult to read, due to several shortages in the use of the English language, and should be rewritten eìwith the help of a mother tongue. In particular, there are terms and sentences that must be replaced by correct ones (es: sentence in lines 41-43 of abstract; "incitant" page 2; sentence in lines 74-75 page 3; is instead of are line 120 page 5; between instead of in, line 186 page 7; verbes used in different times throughtou the materials and methods and results sections; etc etc)

Second, the abstract should be rewritten to evidence in the corect order: background, aim, methods, results, and conclusions, whereas in the current form has two-third of introduction, one conclusive sentence before results (lines 45-47), and results and methods schematically confined in the last lines (47-55, starting from Findings:).

Also the rest of the manuscript suffers from the same problems.

More specifically, in Methods it is difficult to understand if the authors performed "experiment" (as they declare in the abstract) or rather use the already originated metadata (on Asthma study) to analyze existing data aimed to evidence alterations in nasopharyngeal microbiome. The two groups are very disonìmogeneous as to the number (23 vs 82) and this must be justified.

Beside, it seems that of the 23 subjects the authors have saliva from 16 and nasopharyngeal samplkes from 8 (and 8 subjects is a really low number to make significant comparisons!)

Table 1 of the Methods should be moved to the Results.

The quality of the Figures is very porr, they are barely readable and the resolution should be increased.

The figure legends should be more explicative, since in the actual form they are very synthetic and do not allow to understand the meaning; the correspondent text in the manuscript should also display more in deep the results and their meaning.

The discussion is a summary of the results and should instead discuss the obtained data in light of the previous published data, comparing them and highlitinh similarities and differences.

The paragraph "Machine learning" should be moved to Methods, leaving in the Results section only the data obtained and not the description of the method itself.

The conclusion chapter starts with a sentence that has no sense: "demonstrated the significance between"... please rephrase it and try giving a conclusion that put the results in a perspective of importance and possible usage in the treatment of patients.

Also, the bibliography should be updated with the recent papers on the topic (for example, the role of oral microbiome dysbiosis in the COVID19 patients and its relation with severe symptoms was reported in doi: 10.3389/fmicb.2021.687513, and should be included).

Reviewer #2: General comments:

The authors analyze genomic data from saliva and NP samples of patients with and without asthma who were infected with SARS-CoV-2. Their aim was to determine microbial differences, using metagenomic sequencing, between the populations and possibly uncover associations with respiratory exacerbations and diagnostic biomarkers. They acquired the data using NCBI BioProject to identify their study groups and sequenced the samples (23 from SARS-CoV-2-Asthmatic category, 82 from the SARS-CoV-2-non-Asthmatic category). From the data sets, the authors found no significant differences in abundancies between SARS-CoV-2-Asthmatic and SARS-CoV-2-non-Asthmatic types between the salivary and NP microbiomes. Corynebacterium was higher in the SARS-CoV-2-Asthmatic group from the NP samples and Streptococcus showed a more significant difference between the groups in the saliva samples. At the species level, there was a substantial decline in an anti-inflammatory bacterium, Rothia Mucilaginosa, in the SARS-CoV-2-Asthmatic group in the salivary microbiome. Corynebacterium tuberculostearicum, a pathogenic respiratory species, was abundant in the SARS-CoV-2-Asthmatic group in the nasopharyngeal microbiome. Using machine learning algorithms, the study found discriminative features from the frequency table on nasal and saliva microbiomes.

Major Comments:

Although the authors looked at microbial differences between 2 different sample sites, different sexes and different races, which was a strength. However, no conclusions should be made based on this data without baseline microbiomes prior to SARS-CoV-2 infection. The differences between the SARS-CoV-2-Asthmatic and SARS-CoV-2-non-Asthmatic types could be due to the asthma itself, and not the SARS-CoV-2 infection.

Thus, I do not believe that their conclusions are supported by their data.

Minor comments.

Line 70: Change asthmatics to “people with asthma”

Strengthen the argument on why the microbiome of patients with asthma is affected by SARS-CoV-2.

Line 81-83: How does finding elevated expression of ACE2 during COVID-19 infection lead to dysbacteriosis?

Line 318: Was this statistically significant? Were any of the findings statistically significant?

Line 322: The authors indicated that the analysis showed a significant variation yet the p-values are >0.05

Line 350: The authors can’t make the conclusion that the abundance of Streptococcus at the genus level in SARS-CoV-2-Asthmatic samples on salivary microbiome strengthens the proof of an impact on the host respiratory condition with microbial dysbiosis. They don’t have control samples from the Asthma population before SARS-CoV-2 infection. The dysbiosis could stem from asthma itself, not SARS-CoV-2.

In general, grammar should be improved throughout the manuscript.

6. PLOS authors have the option to publish the peer review history of their article (what does this mean?). If published, this will include your full peer review and any attached files.

Reviewer #1: No

Reviewer #2: No

---

## [Author Response · Author response to Decision Letter 0]

28 Jun 2023

Reviewer #1: 

The authors present a study analyzing the composition and potential role of the oro-pharyngeal microbiome in the increased susceptibility of asthmatic subjects to COVID19.

The topic is interesting and sound, however I have several major criticisms.

First, the paper is quite difficult to read, due to several shortages in the use of the English language, and should be rewritten eìwith the help of a mother tongue. 

1. In particular, there are terms and sentences that must be replaced by correct ones (es: sentence in ines 41-43 of abstract;"incitant" page 2; sentence in lines 74-75 page 3; is instead of are line 120 page 5; between instead of in, line 186 page 7; verbes used in different times throughtou the materials and methods and results sections; etc etc)

Response: As per the reviewer's suggestions the above-mentioned changes have been made to the manuscript and the changes have been highlighted. 

2. Second, the abstract should be rewritten to evidence in the corect order: background, aim, methods, results, and conclusions, whereas in the current form has two-third of introduction, one conclusive sentence before results (lines 45-47), and results and methods schematically confined in the last lines (47-55, starting from Findings:).

Also the rest of the manuscript suffers from the same problems. 

Response: The abstract is reordered as suggested by the reviewer. The remaining manuscript sections are modified accordingly.

3. More specifically, in Methods it is difficult to understand if the authors performed "experiment" (as they declare in the abstract) or rather use the already originated metadata (on Asthma study) to analyze existing data aimed to evidence alterations in nasopharyngeal microbiome.

Response: As mentioned in the study, we used the samples from the nasopharyngeal and salivary microbiomes of patients with and without preexisting asthma in response to SARS-COV-2 infection; the BioProject ID: PRJEB51261 has also been mentioned in the manuscript. Further, it has also been mentioned that the dataset files were obtained from the SRA database using the SRA toolbox.

4. The two groups are very disonìmogeneous as to the number (23 vs 82) and this must be justified. Beside, it seems that of the 23 subjects the authors have saliva from 16 and nasopharyngeal samplkes from 8 (and 8 subjects is a really low number to make significant comparisons!)

Response: The number of samples is not defined by the authors. The dataset is made publicly available by the depositors in the SRA under the BioProject ID: PRJEB51261 and is accessed to conduct this experiment. The authors introduce no bias, and all the samples provided in the repository under the particular ID are used for the study. 

5. Table 1 of the Methods should be moved to the Results.

Response: As per the reviewer's suggestions the, Table 1 has been moved to the results part. 

6. The quality of the Figures is very porr, they are barely readable and the resolution should be increased.

Response: The figure quality is enhanced to 300 DPI. Some images might still have less visibility due to their dense representation. 

7. The figure legends should be more explicative, since in the actual form they are very synthetic and do not allow to understand the meaning; the correspondent text in the manuscript should also display more in deep the results and their meaning.

Response: The figure legends are modified per the reviewer's suggestions. 

8. The discussion is a summary of the results and should instead discuss the obtained data in light of the previous published data, compareng them and highlitinh similarities and differences.

Response: No detailed study exists in how we analyzed the samples specific to the dataset (i.e. SARS-CoV-2 affected individuals with preexisting asthmatic condition). So, instead of comparing the findings with existing results, we explained the microbiome markers in this study.

9. The paragraph "Machine learning" should be moved to Methods, leaving in the Results section only the data obtained and not the description of the method itself.

Response: As per the reviewer's suggestions, the description part of the "Machine learning" paragraph has been moved to the methods part, and the results are presented in the same section.

10. The conclusion chapter starts with a sentence that has no sense: "demonstrated the significance between"... please rephrase it and try giving a conclusion that put the results in a perspective of importance and possible usage in the treatment of patients.

Response: The conclusion section is modified per the reviewer's suggestions.

11. Also, the bibliography should be updated with the recent papers on the topic (for example, the role of oral microbiome dysbiosis in the COVID19 patients and its relation with severe symptoms was reported in doi: 10.3389/fmicb.2021.687513, and should be included). 

Response: As per the reviewer's suggestion, the recent papers on the present study are added to the bibliography.

Reviewer #2

Major Comments:

1. Although the authors looked at microbial differences between 2 different sample sites, different sexes and different races, which was a strength. However, no conclusions should be made based on this data without baseline microbiomes prior to SARS-CoV-2 infection. The differences between the SARS-CoV-2-Asthmatic and SARS-CoV-2-non-Asthmatic types could be due to the asthma itself, and not the SARS-CoV-2 infection.Thus, I do not believe that their conclusions are supported by their data.

Minor comments.

1. Line 70: Change asthmatics to "people with asthma"

Response: As per the reviewer's suggestion, the term has been modified and highlighted.

2. Strengthen the argument on why the microbiome of patients with asthma is affected by SARS-CoV-2.

Response: Immunity has been known to play a very important role in the cause of pathogenesis in asthma. Researchers have postulated that asthma patients may be at high risk of being infected with SARS-CoV-2 and experiencing more severe outcomes when infected. People with asthma harbor an altered airway microbiota linked to increased susceptibility to severe illnesses upon viral respiratory infections. However, the microbiomes of patients with asthma during SARS‐CoV‐2 infection have not yet been characterized properly. Consequently, the current study examines the microbial composition in nasopharyngeal and saliva samples from COVID‐19 patients with and without preexisting asthma.

3. Line 81-83: How does finding elevated expression of ACE2 during COVID-19 infection lead to dysbacteriosis?

Response: SARS-CoV-2 infects human cells by using angiotensin-converting enzyme 2 (ACE2) as a receptor, cleaved by transmembrane proteases during host cell infection, thus reducing its activities. ACE2, a homologue of ACE, has been described as a negative regulator of the Renin-Angiotensin System (RAS), alleviating the deleterious actions mediated by Ang II signaling through Ang II receptor type 1 (AT1R) (Kuba et al., 2013). This role is extremely relevant in pathological conditions associated with RAS overactivation, including those related to cardiovascular, renal, and pulmonary systems (Cole-Jeffrey et al., 2015). Besides, ACE2 also displays non-RAS-related roles linked with neutral amino acids transport and gut homeostasis; elevated ACE2 expression or function are potentially promoters of intestinal dysbiosis (Perlot and Penninger, 2013). This is aligned with the gastrointestinal (GI) symptoms, such as nausea and diarrhea, reported in COVID-19 patients, suggesting an impact on the gastrointestinal-enteric system (Kotfis and Skonieczna-Zydecka, 2020).

References:

Kuba, K., Imai, Y., & Penninger, J. M. (2013). Multiple functions of angiotensin-converting enzyme 2 and its relevance in cardiovascular diseases. Circulation Journal, 77(2), 301-308.

Cole-Jeffrey, C. T., Liu, M., Katovich, M. J., Raizada, M. K., & Shenoy, V. (2015). ACE2 and microbiota: emerging targets for cardiopulmonary disease therapy. Journal of cardiovascular pharmacology, 66(6), 540.

Perlot, T., & Penninger, J. M. (2013). ACE2–From the renin–angiotensin system to gut microbiota and malnutrition. Microbes and infection, 15(13), 866-873.

Kotfis, K., & Skonieczna-Żydecka, K. (2020). COVID-19: gastrointestinal symptoms and potential sources of 2019-nCoV transmission. Anaesthesiol Intensive Ther 2020; 40157.

4. Line 318: Was this statistically significant? Were any of the findings statistically significant?

Response: Yes. The statistical significance between the subgroups is explained through alpha, beta diversity analysis, and LDA scores by cladogram analysis.

5. Line 322: The authors indicated that the analysis showed a significant variation yet the p-values are >0.05

Response: The findings reported in the manuscript conclude that there is little significance between the asthmatic and non-asthmatic SARS-CoV-2 infected groups identified in both nasal and oral microbiome samples.

6. Line 350: The authors can't make the conclusion that the abundance of Streptococcus at the genus level in SARS-CoV-2-Asthmatic samples on salivary microbiome strengthens the proof of an impact on the host respiratory condition with microbial dysbiosis. They don't have control samples from the Asthma population before SARS-CoV-2 infection. The dysbiosis could stem from asthma itself, not SARS-CoV-2.

Response: Our conclusion does not emphasize the abundance of Streptococcus as the proof but reports the findings as the discriminating evidence between Asthmatic/non-Asthmatic cases having SARS-CoV-2 condition. It is similar to an example of comparing the high-risk HPV vs low-risk HPV cases to understand the biological differences, thereby could treat with effective medications among the subgroups. 

7. In general, grammar should be improved throughout the manuscript.

Response: Based on the reviewer's suggestion, the manuscript's grammar has been improvised.

---

## [Decision Letter · Decision Letter 1]

17 Jul 2023

PONE-D-23-05664R1Airway and Oral Microbiome Profiling of SARS-CoV-2 Infected Asthmatic and Non-Asthmatic Cases Revealing Alterations – A Pulmonary Microbial InvestigationPLOS ONE

Dear Dr. El Allali,

Thank you for submitting your manuscript to PLOS ONE. After careful consideration, we feel that it has merit but does not fully meet PLOS ONE’s publication criteria as it currently stands. Therefore, we invite you to submit a revised version of the manuscript that addresses the points raised during the review process.

Thank you for submitting your work to PLoS One. The two original reviewers have reviewed and commented on your manuscript. Reviewer one is in favor or acceptance while reviewer 2 has recommended a minor revision. As noted in my first decision, I agree with reviewer 2 that it is important to acknowledge the limitations of the current study. This change will be critical in the assessment of your revised manuscript. ================================

We look forward to receiving your revised manuscript.

Kind regards,

Robert P Smith

Academic Editor

PLOS ONE

Journal Requirements:

Reviewers' comments:

Reviewer's Responses to Questions

**Comments to the Author**

1. If the authors have adequately addressed your comments raised in a previous round of review and you feel that this manuscript is now acceptable for publication, you may indicate that here to bypass the “Comments to the Author” section, enter your conflict of interest statement in the “Confidential to Editor” section, and submit your "Accept" recommendation.

Reviewer #1: All comments have been addressed

Reviewer #2: (No Response)

2. Is the manuscript technically sound, and do the data support the conclusions?

Reviewer #1: Yes

Reviewer #2: Partly

3. Has the statistical analysis been performed appropriately and rigorously? 

Reviewer #1: Yes

Reviewer #2: Yes

4. Have the authors made all data underlying the findings in their manuscript fully available?

Reviewer #1: Yes

Reviewer #2: Yes

5. Is the manuscript presented in an intelligible fashion and written in standard English?

Reviewer #1: Yes

Reviewer #2: Yes

6. Review Comments to the Author

Reviewer #1: The authors have adequately addressed to the comments raised by both Reviewers in a previous round of review.

Consequently, the revised manuscript is considerably improved and I think that it can be accepted.

Reviewer #2: Thank you for revising the manuscript. My original concern has not been addressed. You still have not shown the microbial data from uninfected samples. If that is not feasible to do, then please address that in the discussion. Add a "limitations" paragraph where you address some of the limitations of the study such as the low number of some of the sample types as well as the lack of uninfected controls.

7. PLOS authors have the option to publish the peer review history of their article (what does this mean?). If published, this will include your full peer review and any attached files.

Reviewer #1: No

Reviewer #2: No

---

## [Author Response · Author response to Decision Letter 1]

20 Jul 2023

Reviewer #2: 

Comment: Thank you for revising the manuscript. My original concern has not been addressed. You still have not shown the microbial data from uninfected samples. If that is not feasible to do, then please address that in the discussion. Add a "limitations" paragraph where you address some of the limitations of the study such as the low number of some of the sample types as well as the lack of uninfected controls.

Response: Based on reviewer suggestion, we have added a separate section as “Limitation of the present study” after the conclusion.

---

## [Decision Letter · Decision Letter 2]

28 Jul 2023

Airway and Oral Microbiome Profiling of SARS-CoV-2 Infected Asthmatic and Non-Asthmatic Cases Revealing Alterations – A Pulmonary Microbial Investigation

PONE-D-23-05664R2

Dear Dr. El Allali,

We’re pleased to inform you that your manuscript has been judged scientifically suitable for publication and will be formally accepted for publication once it meets all outstanding technical requirements.

Kind regards,

Robert P Smith

Academic Editor

PLOS ONE

Additional Editor Comments (optional):

Reviewers' comments:

Reviewer's Responses to Questions

**Comments to the Author**

1. If the authors have adequately addressed your comments raised in a previous round of review and you feel that this manuscript is now acceptable for publication, you may indicate that here to bypass the “Comments to the Author” section, enter your conflict of interest statement in the “Confidential to Editor” section, and submit your "Accept" recommendation.

Reviewer #2: All comments have been addressed

2. Is the manuscript technically sound, and do the data support the conclusions?

Reviewer #2: (No Response)

3. Has the statistical analysis been performed appropriately and rigorously? 

Reviewer #2: (No Response)

4. Have the authors made all data underlying the findings in their manuscript fully available?

Reviewer #2: (No Response)

5. Is the manuscript presented in an intelligible fashion and written in standard English?

Reviewer #2: (No Response)

6. Review Comments to the Author

Reviewer #2: (No Response)

7. PLOS authors have the option to publish the peer review history of their article (what does this mean?). If published, this will include your full peer review and any attached files.

Reviewer #2: No

---

## [Editor Report · Acceptance letter]

9 Aug 2023

PONE-D-23-05664R2 

Airway and Oral Microbiome Profiling of SARS-CoV-2 Infected Asthma and Non-Asthma Cases Revealing Alterations – A Pulmonary Microbial Investigation 

Dear Dr. El Allali:

I'm pleased to inform you that your manuscript has been deemed suitable for publication in PLOS ONE. Congratulations! Your manuscript is now with our production department. 

Kind regards, 

on behalf of

Dr. Robert P Smith 

Academic Editor

PLOS ONE